# The validity and reliability of wearable devices for the measurement of vertical oscillation for running

**Craig P. Smith** [ORCID]*, **Elliott Fullerton, Liam Walton, Emelia Funnell**¤, **Dimitrios Pantazis, Heinz Lugo**

INCUS Performance Ltd., Loughborough, United Kingdom

¤ Current address: Gymshark, Solihull, United Kingdom
* c.smith@incusperformance.com

**Data Availability Statement:** All relevant data are within the paper and its Supporting Information files.

## Abstract

Wearable devices are a popular training tool to measure biomechanical performance indicators during running, including vertical oscillation (VO). VO is a contributing factor in running economy and injury risk, therefore VO feedback can have a positive impact on running performance. The validity and reliability of the VO measurements from wearable devices is crucial for them to be an effective training tool. The aims of this study were to test the validity and reliability of VO measurements from wearable devices against video analysis of a single trunk marker. Four wearable devices were compared: the INCUS NOVA, Garmin Heart Rate Monitor-Pro (HRM), Garmin Running Dynamics Pod (RDP), and Stryd Running Power Meter Footpod (Footpod). Fifteen participants completed treadmill running at five different self-selected speeds for one minute at each speed. Each speed interval was completed twice. VO was recorded simultaneously by video and the wearables devices. There was significant effect of measurement method on VO (p < 0.001), with the NOVA and Footpod underestimating VO compared to video analysis, while the HRM and RDP overestimated. Although there were significant differences in the average VO values, all devices were significantly correlated with the video analysis (R > = 0.51, p < 0.001). Significant agreement between repeated VO measurements for all devices, revealed the devices to be reliable (ICC > = 0.948, p < 0.001). There was also significant agreement for VO measurements between each device and the video analysis (ICC > = 0.731, p < = 0.001), therefore validating the devices for VO measurement during running. These results demonstrate that wearable devices are valid and reliable tools to detect changes in VO during running. However, VO measurements varied significantly between the different wearables tested and this should be considered when comparing VO values between devices.

## Introduction

The availability and popularity of wearable sports technology for running has grown extensively in recent years [1]. These devices provide users with feedback about a variety of

**Funding:** CPS, EF, LW, EF, DP, and HL were funded by Innovate UK (project no. 00106514). https://www.ukri.org/councils/innovate-uk/?_ga=2. 89826907.1149472773.1647884579-1155892482. 1640269449. The funders had no role in study design, data collection and analysis, decision to publish, or preparation of the manuscript.

**Competing interests:** I have read the journal's policy and the authors of this manuscript have the following competing interests: The INCUS NOVA wearable device used in the research article is a license product of INCUS Performance Ltd. CPS, EF, LW, EF, DP, and HL were employees of INCUS Performance Ltd. at the time the research was completed. This does not alter our adherence to PLOS ONE policies on sharing data and materials.

biomechanical information, including running specific metrics such as speed and cadence. The affordability and portable nature of wearable devices make them an attractive method for measuring biomechanical features of running outside of the laboratory for runners [2], researchers [3] and clinicians alike [4].

A running specific metric provided by several wearable devices is vertical oscillation (VO). VO is the vertical displacement of the body during each stride measured at the centre of mass (COM) [5], or proxy positions such as the pelvis [6]. VO has been linked to running economy and injury prevention, with smaller VO of the trunk associated with improved running economy [7, 8] and a reduction in lower-limb injury risk factors such as vertical loading rate [9]. Running technique can be adapted to alter VO [9–11], such as increasing cadence to reduce VO [12]. Providing a runner with real-time visual and auditory feedback that indicates when their vertical oscillation is above or below a target level can allow a runner to manipulate their vertical oscillation as required [13]. Therefore, wearable devices have the potential to provide an accessible method for runners and coaches to obtain and utilise VO feedback for performance gains.

There are a variety of wearable devices currently on the market which provide VO measurements for running. These devices commonly utilise an inertial measurement unit (IMU) to record body movement and derive a variety of biomechanical features during running. The position of recording varies between wearables, with some devices positioned on the trunk at the xiphoid process (Garmin Heart Rate Monitors), C7 vertebrae (INCUS NOVA), waistband (Garmin Running Dynamics Pod), or on the dorsum of the foot (Stryd Running Power Meter Footpod). The validity and reliability of VO measurements from wearable devices is essential to determine whether the device can detect changes in VO or whether changes are the result of measurement errors. However, few studies have focused on validating wearable devices for VO measurements. VO recorded from Garmin heart-rate monitors with built in accelerometers (HRM) have been compared to a video analysis method and found to be highly agreeable [14, 15], as well as reliable between repeated measures [14]. The manufacturers of the Stryd Running Power Meter Footpod (Footpod) report that the device measures COM VO with a small average error of 3% when compared to a ground reaction forces method for deriving COM VO [16]. These findings provide some evidence that wearable technology can be a valid and reliable tool for measuring VO. However, the validity and reliability of VO for other devices, such as the INCUS NOVA (NOVA) and Garmin Running Dynamics Pod (RDP) has not been reported. Furthermore, it is not understood how VO measurements from different devices compare, especially given they record at different locations on the body.

The aim of this study was to test whether wearable devices are reliable and valid tools for the measurement of VO during running by comparing VO measurements from four wearable devices (NOVA, HRM, RDP, and Footpod) to video analysis of a single trunk-based marker. Based on prior research [14–16], it is hypothesised that the wearable devices will provide valid and reliable VO measurements when compared to video analysis. However, because of the difference in body locations between devices, it is hypothesised that the VO measurements will differ between devices, with the device in closest proximity to the trunk marker (NOVA) having the most accurate VO measurements when compared to the video analysis measurements.

## Materials and methods

### Participants

Fifteen active runners (run for at least 1 hour per week) without any injury in the last 6 months were recruited (7 females, mean ±SD age = 26.4yrs ±5.5, height = 174.4cm ±9.4, weight = 71.1kg ±9.3). All participants gave written informed consent, and the experiment was

conducted in accordance with the Declaration of Helsinki. Ethical approval for this research was obtained from the Loughborough University Ethics Review Committee (ERSC22_27).

## Apparatus

The participants ran indoor on a motorised treadmill (NordicTrack T8.5S, NordicTrack, Utah, USA) in their usual running footwear at a 1% incline with a fan to mimic outdoor running [17] (Fig 1). During running, VO was measured using video analysis and four wearable devices designed to measure VO during running: INCUS NOVA (INUCS Performance Ltd., Loughborough, UK), Garmin HRM (Garmin Ltd., Southampton, UK), Garmin RDP (Garmin Ltd., Southampton, UK), and Stryd Footpod (Stryd, Colorado, US). The wearable devices were worn, and data recorded, as per their instructions. The NOVA was worn in a purpose-built harness positioned towards the top of the spine (C7 vertebrae) and paired with the INCUS mobile application to start and stop data recording. The HRM was fitted using the accompanying chest strap, with the device located on the xiphoid process and paired with the Garmin

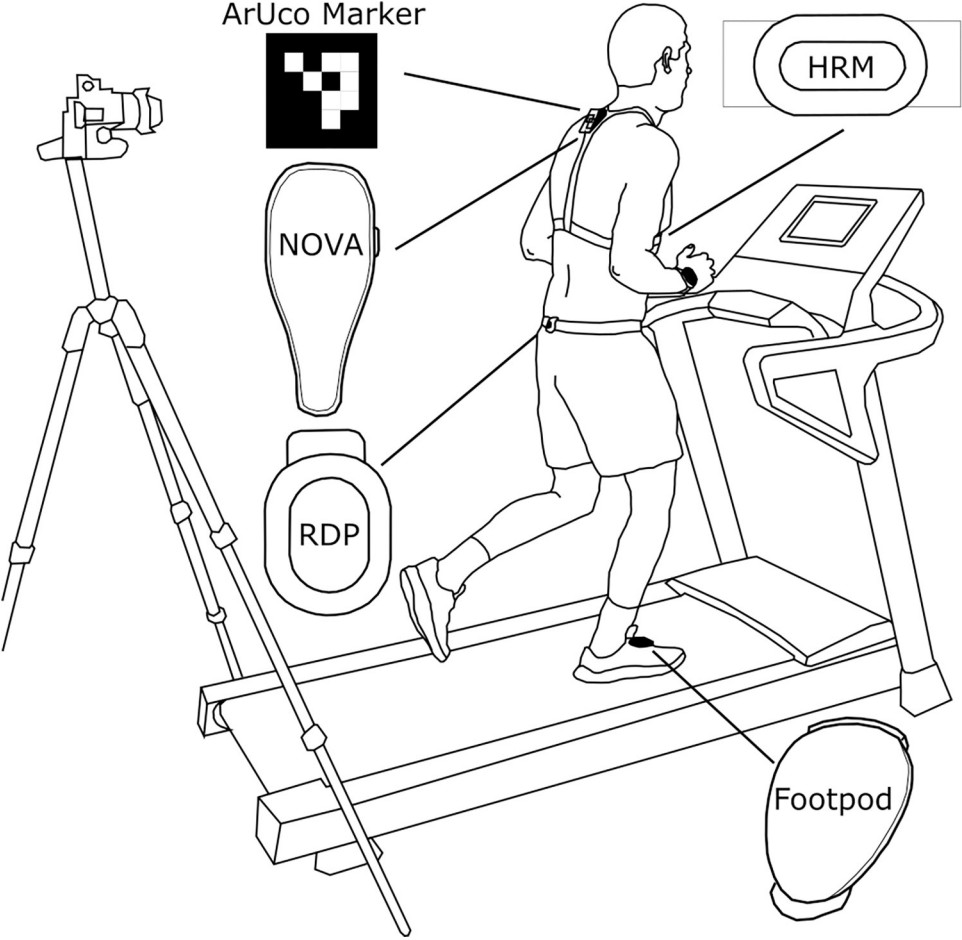

**Fig 1. Experimental setup.** The illustration shows a participant aboard the treadmill and the positioning of the wearable devices on the body. The INCUS NOVA (NOVA) was worn in a 'T-Strap', with the device positioned at C7 vertebrae. The ArUco marker was fixed to the NOVA and video recorded from the rear. The Garmin HRM-Pro chest strap (HRM) was worn with the device positioned at the xiphoid process. The Garmin Running Dynamics Pod (RDP) was clipped to the waistband, aligned with the sagittal plane. The Stryd Running Power Meter Footpod (Footpod) was clipped to the laces of the right trainer. Depictions of the wearable devices are for illustrative purposes only.

Forerunner 945 watch (Garmin Ltd., Southampton, UK) worn by the participant on their right wrist. The RDP was fitted to the rear of the participants waistband aligned with the sagittal plane and paired with a separate Garmin Forerunner 245 watch (Garmin Ltd., Southampton, UK) worn on the participants left wrist. Both watches were set to treadmill mode and the recording of data from both Garmin devices was started/stopped via their corresponding watches. The Footpod was clipped to the lower two laces of the participants right trainer and controlled via the Stryd mobile application.

A video analysis system was used to provide a reference to test the validity of the VO measurements of the wearable devices. An ArUco marker, a 5 x 5cm square with a black border and an inner black and white binary matrix [18], was fixed to the NOVA device and a digital single-lens reflex camera (Panasonic Lumix DMC-FZ330 Digital, Panasonic, UK) was positioned on a tripod to the rear of the treadmill at the same height as the marker. The marker was video recorded at 200 FPS in 4K. It is possible to accurately measure movements of an ArUco marker <2.2m/s using video recording [19]. Vertical trunk movement during treadmill running was not expected to surpass this velocity limit under the conditions of this study and therefore the video recording of an ArUco marker was deemed a suitable method for measuring VO.

## Protocol

Participants self-selected a range of five preferred running speeds of 1km/h increments (e.g., 8–12 km/h) during a five-minute familiarisation period on the treadmill. The participants ran for one-minute intervals at each speed, with a minute of slow walking (1 km/h) between each interval. This was completed twice in two blocks, with three minutes of slow walking between the blocks (Fig 2). Therefore, a total of 10 running intervals (2 blocks x 5 speeds) were completed by each participant. The order of the running speeds was randomised (pseudorandom number generator) within each block and for each participant. Data recording from the wearable devices were started by the researcher one minute prior to the beginning of the first running interval and recorded continuously throughout the protocol. Video recording of the marker was collected by a researcher during each running interval.

## Data and statistical analysis

The data recording from the NOVA was downloaded via the INCUS mobile application. The data from the watches were downloaded using the Garmin connect software for the HRM and RDP. Footpod data was downloaded using the Stryd Power Center software.

Video recordings of the marker positioned on the NOVA device allowed for automatic detection of the trunk (C7 vertebrae) and NOVA's position throughout each trial. ArUco marker detection was achieved using the open-source library OpenCV [20]. First, using a checkerboard calibration recording [21], any distortion in the camera image was removed from each frame. Each frame was then converted to a grayscale image to achieve accurate marker detection and shorter processing times. The pixel co-ordinates of each corner of the marker were detected and the location of the centre of the marker derived. Using the marker centre location, a pixel to centimetre ratio was calculated using the arclength method to measure the marker perimeter which has a known size (5x5cm). The difference between maximum and minimum position of the marker in the vertical axis was calculated for each stride and converted to centimetres using the pixel to centimetre ratio to derive vertical oscillation.

Time synchronisation of VO values between measurement methods was achieved by finding the start and end time of each of the ten running intervals within the VO values recorded for each method. For each device the starts of the intervals were clearly visible as a rapid rise in

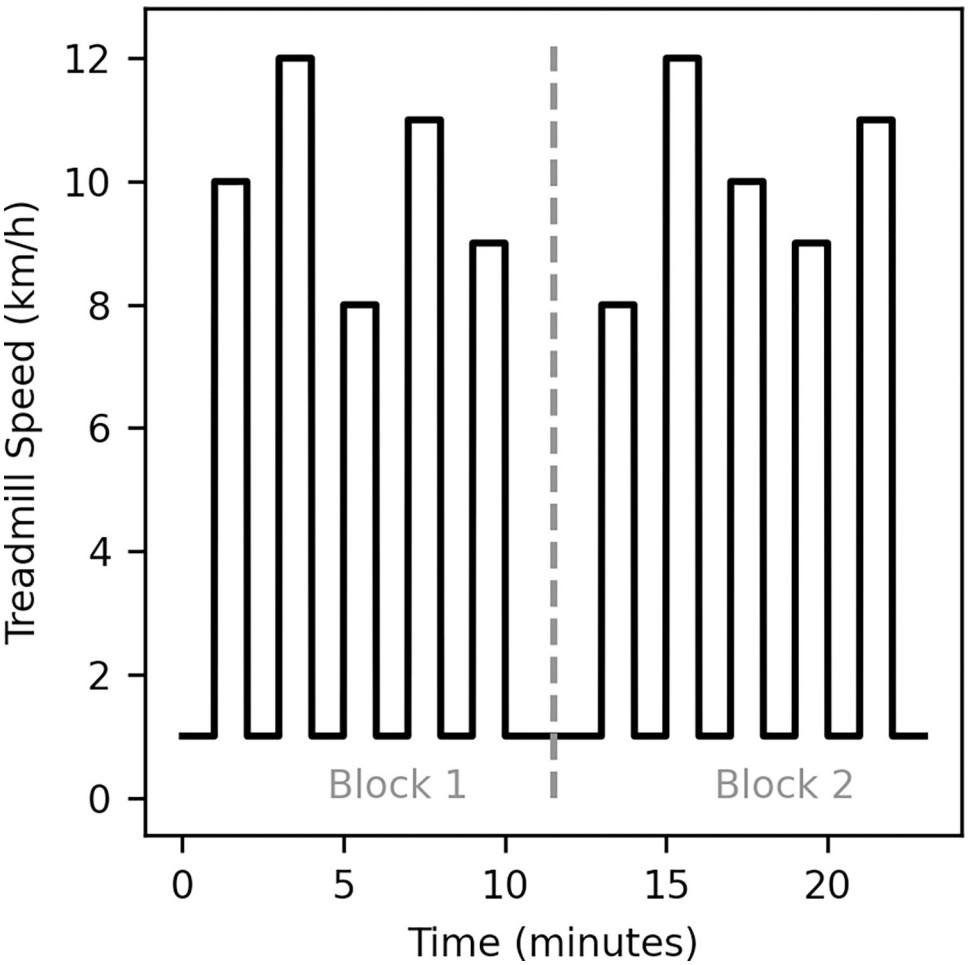

**Fig 2. Running intervals.** An example of the running interval profile for a participant with a preferred running speed range of 8–12 km/h. Participants ran for one minute at each selected speed with a one-minute break (walking at 1 km/h) between each running interval. This was completed twice (Block 1 & 2) with the speed order randomised within each block and a three-minute walking period between the blocks.

VO, which remained elevated until the end of the interval when VO would then rapidly fall. Therefore, the intervals were defined by the rise and fall of VO above and below the mean VO across the whole protocol for each measurement method. For each interval, the mean and standard deviation of VO was calculated for the middle 30 seconds, and the corresponding running intervals were compared between methods.

Differences in average VO (bias and 95% limits of agreement) between the video analysis and the other methods was calculated and a 1 x 5 repeated measure analysis of variance (ANOVA) was used to test for a main effect of measurement method on VO. Post-hoc paired T-tests were then calculated to test for differences between each of the methods, therefore ten comparisons in total. To reduce the likelihood of Type 1 errors when making multiple comparisons, the alpha level for the paired T-tests was Bonferroni corrected, therefore divided by the number of comparisons (i.e. alpha = 0.05/10). To determine the strength of the relationship between device VO and the video analysis method, repeated measures correlations [22] were calculated between the video analysis VO and each of the wearable devices across all running interval speeds. To determine the reliability of each method, VO measurements during the

participants mid-range selected running speed interval in the first block was compared to the second block using Intraclass Correlation Coefficients ($ICC_{3,1}$), and the standard error of measurement (SEM) was calculated. To determine the validity of the devices, the VO measurements recorded for the participants mid-range selected running speed interval were averaged across the two blocks of trials and $ICC_{3,1}$ were calculated to test the agreement between the devices and video analysis. Statistical analyses were carried out using Python 3.0.0 library Pingouin 0.5.0, with the alpha level set at 0.05.

## Results

The mean (±SD) preferred running speed range was 8.9–12.9 km/h ±1.2. The average VO measurements for each device across all running intervals are shown in Fig 3. There was a significant difference in average VO between the methods of measurement ($F_{(4, 56)} = 39.70$, $p < 0.001$). Post-hoc pairwise comparisons between devices (Bonferroni corrected alpha level = 0.005) were all significant ($t_{(14)} > = 4.32$, $p < 0.001$) other than between HRM and RDP ($t_{(14)} = 0.40$, $p = 0.692$), and NOVA and Footpod ($t_{(14)} = 3.16$, $p = 0.007$).

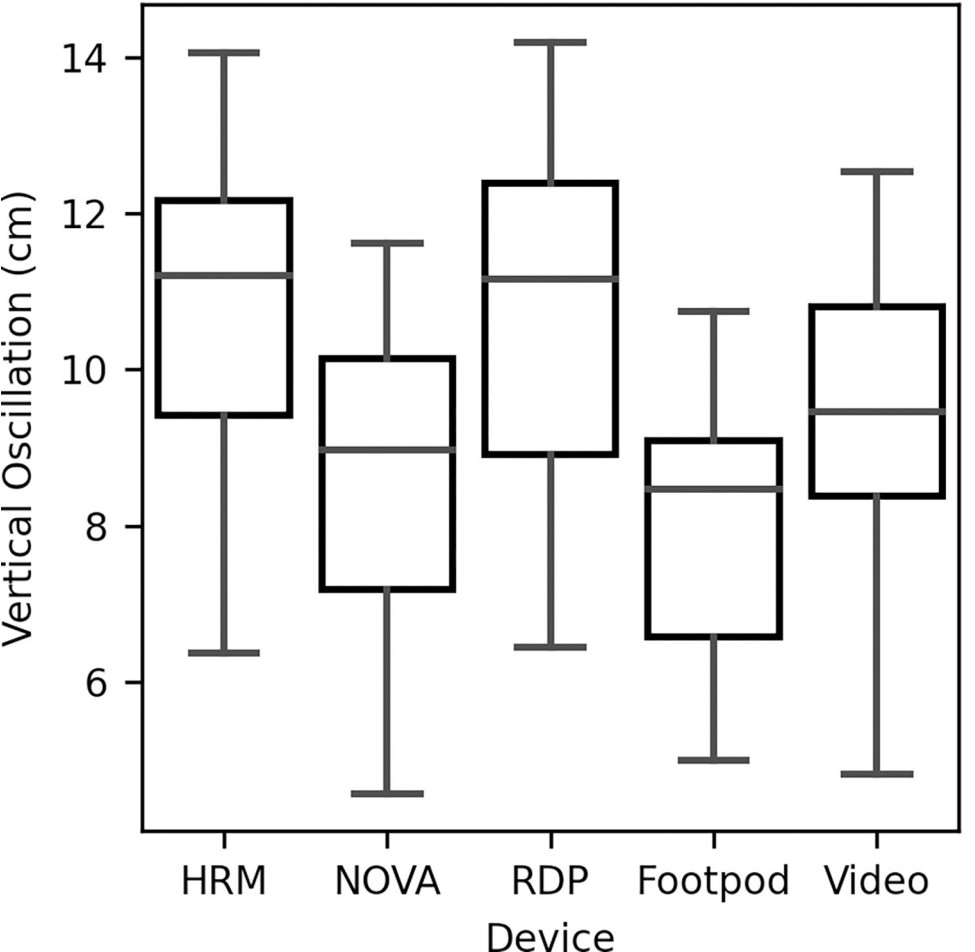

**Fig 3. Average vertical oscillation for each method.** The box plot shows the distribution of VO values for each device. Median VO and quartiles 1 (Q1) and 3 (Q3) are show by the box, while lower and upper bars are Q1–1.5*Inter-Quartile Range and Q3 + 1.5*Inter-Quartile Range, respectively. There were no outliers above or below the bars for any method.

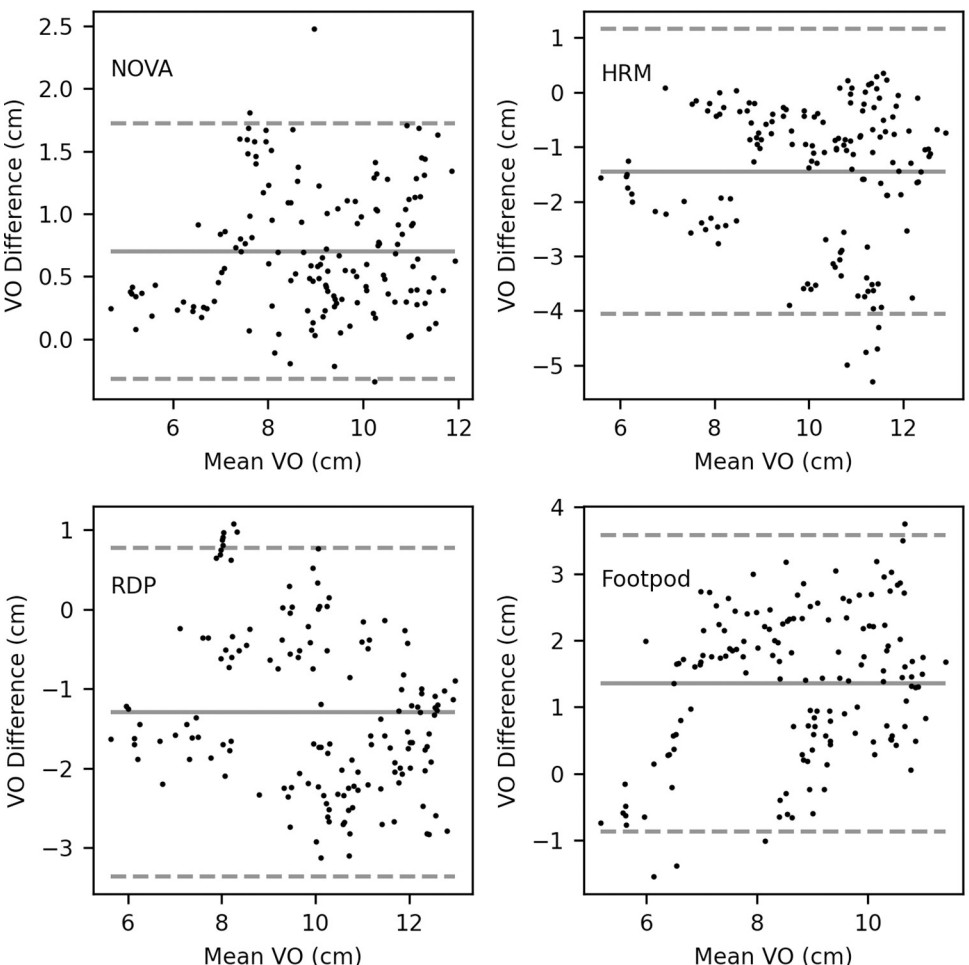

**Fig 4. Bland-Altman plots between video analysis vertical oscillation and wearable devices.** Bland-Altman plot for video analysis vertical oscillation values compared to INCUS NOVA (top left), Garmin HRM-Pro chest strap (HRM, top right), Garmin Running Dynamics Pod (RDP, bottom left), and Stryd Running Power Meter Footpod (Footpod, bottom right). Mean bias is indicated by the solid line. Dashed lines indicate 95% Limits of Agreement. All running intervals (n = 10) and participants (n = 15) were included.

The Bland-Altman plots (Fig 4), along with the pairwise comparisons reveal that the HRM (10.8cm ±1.5) and RDP (10.7cm ±2.1) overestimated VO compared to the video analysis (9.4cm ±1.8), while the NOVA (8.7cm ±1.7) and Footpod (8.0cm ±1.5) underestimated.

Although, there was a difference in the average VO between the devices and video analysis, the NOVA (R = 0.84, p < 0.001), HRM (R = 0.73, p < 0.001), RDP (R = 0.80, p < 0.001), and Footpod (R = 0.51, p < 0.001) were significantly correlated with the video analysis values (Fig 5).

To test the reliability of the VO measurements, the VO values for the participants mid-range running speed interval were compared between the first (block 1) and repeated (block 2) measurement. All devices had significant reliability between the repeated measurements ($ICC_{3,1} > = 0.928$, $F_{(14,14)} > = 26.87$, p < 0.001) and standard error of measurements < = 0.5cm (Table 1).

The validity of each device when compared to the video analysis was also tested for the mid-range running interval (Table 2). There was significant agreement between all the devices and the video analysis method ($ICC_{3,1} > = 0.731$, $F_{(14,14)} > = 6.45$, p < = 0.001).

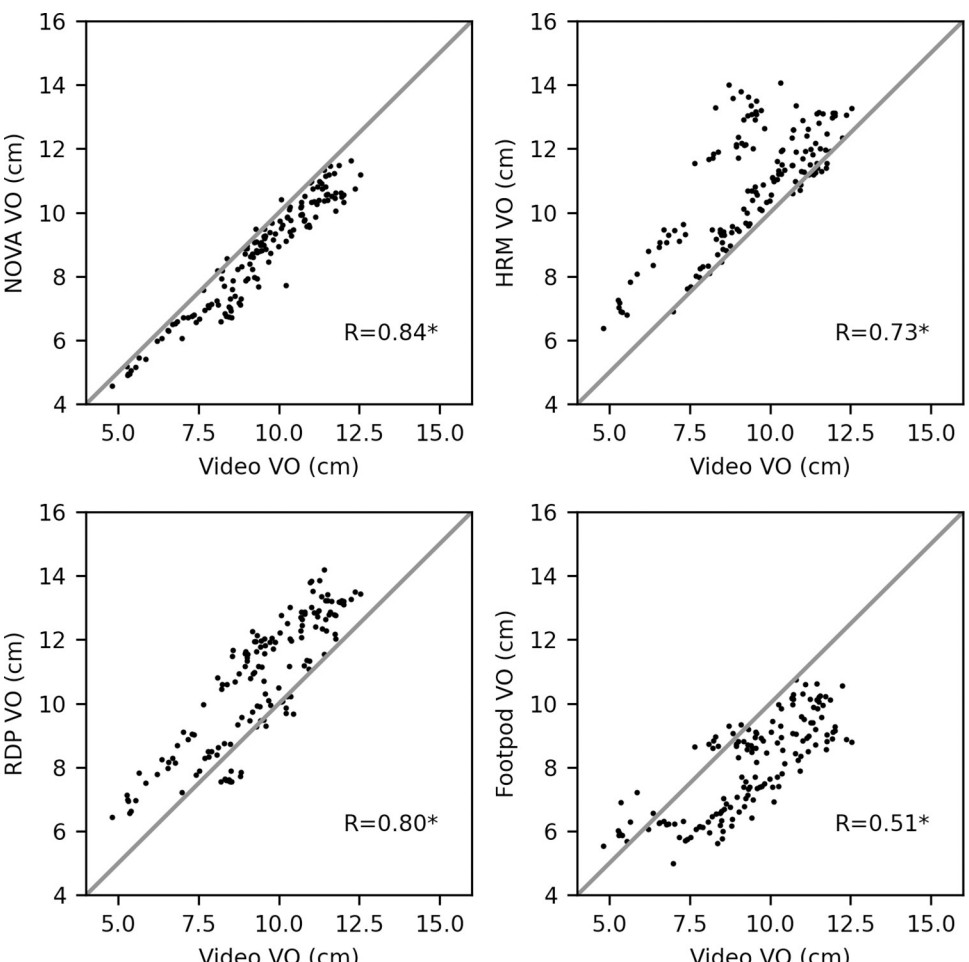

**Fig 5. Video analysis vertical oscillation versus wearable devices.** Scatter plots of the video analysis vertical oscillation measurements (VO) versus VO values from four wearable devices; the INCUS NOVA (top left), Garmin HRM-Pro chest strap (top right), Garmin Running Dynamics Pod positioned on the waistband (bottom left), and the Stryd Running Power Meter Footpod (bottom right). All running intervals (n = 10) and participants (n = 15) are included. The diagonal line represents the line of unity for the video analysis measurements. Repeated measures correlation R values between the video analysis and devices are shown (*p < 0.05).

**Table 1. Reliability between repeated vertical oscillation measurements.**

|  | Block 1 Vertical Oscillation (cm) | Block 2 Vertical Oscillation (cm) | ICC [95% CI] | SEM (cm) |
|---|---|---|---|---|
| **Video Analysis** | 9.5 +/- 1.9 | 9.5 +/- 1.8 | 0.928 [0.80, 0.98]* | 0.5 |
| **INCUS NOVA** | 8.7 +/- 1.8 | 8.8 +/- 1.8 | 0.956 [0.87, 0.98]* | 0.4 |
| **Garmin HRM-Pro** | 10.9 +/-1.8 | 11.1 +/-1.8 | 0.948 [0.85, 0.98]* | 0.4 |
| **Garmin RDP** | 10.7 +/- 2.2 | 10.9 +/- 2.1 | 0.968 [0.91, 0.99]* | 0.4 |
| **Stryd Footpod** | 8.2 +/- 1.5 | 8.2 +/- 1.4 | 0.954 [0.87, 0.98]* | 0.3 |

Mean (±SD) vertical oscillation for the mid-range running speed interval across all participants for the first and second block of trials. ICCs show agreement between the blocks for each method (*p < 0.05). Standard Error of Measurement (SEM) indicates the amount of variability between the repeated measures due to measurement error.

**Table 2. Validity of vertical oscillation measurements from wearable devices compared to video analysis.**

| | Vertical Oscillation (cm) | ICC [95% CI] | Bias (cm) | 95% Limits of Agreement |
|---|---|---|---|---|
| **Video Analysis** | 9.5 +/- 1.8 | | | |
| **INCUS NOVA** | 8.8 +/- 1.7 | 0.963 [0.89, 0.99]* | 0.7 | [-0.3, 1.6] |
| **Garmin HRM-Pro** | 11.0 +/-1.8 | 0.745 [0.39, 0.91]* | -1.5 | [-4.1, 1.1] |
| **Garmin RDP** | 10.8 +/- 2.1 | 0.858 [0.63, 0.95]* | -1.3 | [-3.4, 0.8] |
| **Stryd Footpod** | 8.2 +/- 1.4 | 0.731 [0.37, 0.90]* | 1.3 | [-1.1, 3.7] |

Mean (±SD) vertical oscillation across all participants for the mid-range speed interval. Intraclass correlation coefficients (ICCs) between video analysis and the devices; INCUS NOVA, Garmin HRM-Pro, Garmin Running Dynamics Pod (RDP), and Stryd Running Power Meter Footpod (Footpod) (*$p < 0.05$). Mean bias (±SD) and 95% Limits of Agreement between video analysis values and the devices are shown.

## Discussion

The agreement between the wearable devices and the video analysis reference, along with the high reliability values between repeated measures, indicate that the wearable devices are valid and reliable tools for measuring VO of the trunk during running. As hypothesised, the NOVA measurements had the highest agreement and lowest average bias compared to the video analysis. However, the absolute VO values differed between the devices, with the NOVA and Footpod underestimating VO compared to the video analysis, while the RDP and HRM overestimated.

All four wearable devices had VO measurements which significantly agreed with the video analysis. However, the strength of the agreement varied between devices. Furthermore, the absolute VO values differed significantly between devices. The largest difference was between the Footpod and HRM, with the average Footpod VO 26% lower than the HRM. Compared to the video analysis, the NOVA had the highest correlation and ICC values of the four devices ($R = 0.84$, ICC = 0.96), as well as the smallest average bias (0.7cm). The video analysis measured the vertical movement of a marker fixed to the NOVA, therefore measuring VO at the C7 vertebrae. In contrast, the HRM recorded from the xiphoid process, the RDP recorded from the rear of the waistband, and the Footpod from the foot. When compared to video analysis of a marker fixed to the HRM, the HRM has been found to have strong correlation coefficients (ICC > = 0.96) and minimal bias (< = 0.3cm) when compared with video analysis measurements [14, 15], similar to the results for the NOVA in this study. This suggests that a potential reason for the differences in VO found between the NOVA and HRM is that although the devices are measuring VO referenced to the location of the device, real differences in VO between the measurement locations is the explanation for the difference found between these devices. However, on average there was little difference in VO measurements between the HRM and RDP, although these devices record from contrasting trunk locations. This suggests recording location may not be the sole contributor to differences between the trunk-based devices and that the differences are likely due to a combination of both location and the device itself. Further research comparing device VO to video analysis at each device location will help to understand if biomechanical factors contribute to VO measurement differences when recording at different locations on the trunk.

Although positioned on the foot, the Footpod reports to measure VO of COM [16]. VO of the COM is commonly measured using 3D motion capture and a segmental model of the body is applied to locate COM displacements during running [23]. Measuring VO of COM with either video analysis of a single marker [24] or a single IMU [15, 25] has proven difficult, with both methods overestimating COM VO. A linear correction of IMU VO to infer COM VO has been proposed, although this method is susceptible to overfitting on the sample tested and

requires validation in different cohorts [15]. The overestimation of COM VO when measured by a single marker or IMU on the trunk may explain why the trunk located devices had higher VO values compared to the Footpod which indirectly measures COM VO from measurements taken at the foot.

Although further research is required to understand the mechanisms for the VO differences between devices, this finding demonstrates that caution must be taken when using devices interchangeably. This is an important consideration for users, who may discover significant changes in their VO values when moving from one device to another. An artificial increase in VO measurements could lead a user to unnecessarily adapt their running technique to reduce their VO (e.g., increase cadence) with a negative impact on overall performance. On the other hand, an artificial decrease in VO measurements could result in the user incorrectly believing their VO has improved, preventing them from benefiting from improved running economy [8] and reduced injury risk factors [9] associated with an actual reduction in VO. However, when interpreting the VO feedback from a single device in isolation, this study has found that wearable devices can provide a valid and reliable method for the measurement of VO, which is important for user confidence. The ability to measure VO via a wearable device has the benefits of being unobtrusive and affordable compared to the traditional method of video analysis. Therefore, wearable devices provide a broader range of runners the opportunity to incorporate VO feedback into their training.

In this study, VO was measured during treadmill running. Running on a treadmill compared to overground running could potentially increase VO due to flexion in the treadmill running surface and should be considered when applying the results of treadmill VO studies to outdoor running. Another external factor known to effect VO is running footwear, with evidence that running barefoot reduces VO compared to shod running [26]. In this study, participants wore their own choice of running footwear, therefore footwear type was not controlled for. However, the effect of footwear type on VO was likely minimal considering the effect of barefoot running has been reported to be a 7% reduction in VO [26].

## Conclusions

Wearable devices provide a valid and reliable method for measuring changes in VO during running when compared to a video analysis method. Therefore, such devices give runners an accessible option to track changes in their VO with potential performance and injury related benefits. However, absolute VO values differ between devices, therefore caution must be taken when using devices interchangeably for VO measurements.

## Supporting information

**S1 Dataset.**
(CSV)

## Acknowledgments

The authors would like to thank Emily Codd and Spencer Patmore for their assistance in data collection.

## Author Contributions

**Conceptualization:** Craig P. Smith, Elliott Fullerton, Liam Walton, Emelia Funnell, Heinz Lugo.

**Data curation:** Craig P. Smith, Elliott Fullerton, Liam Walton, Emelia Funnell.

**Formal analysis:** Craig P. Smith, Elliott Fullerton, Heinz Lugo.

**Funding acquisition:** Dimitrios Pantazis, Heinz Lugo.

**Investigation:** Craig P. Smith, Elliott Fullerton, Emelia Funnell.

**Methodology:** Craig P. Smith, Elliott Fullerton, Liam Walton, Emelia Funnell, Heinz Lugo.

**Project administration:** Craig P. Smith, Liam Walton, Heinz Lugo.

**Supervision:** Dimitrios Pantazis, Heinz Lugo.

**Validation:** Craig P. Smith.

**Visualization:** Craig P. Smith.

**Writing – original draft:** Craig P. Smith.

**Writing – review & editing:** Craig P. Smith, Elliott Fullerton, Liam Walton, Dimitrios Pantazis, Heinz Lugo.

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
