## [Decision Letter · Decision Letter 0]

25 Aug 2022

PONE-D-22-11731The Validity and Reliability of Vertical Oscillation Measured by the INCUS NOVA Wearable Device for Running.PLOS ONE

Dear Dr. Smith,

Thank you for submitting your manuscript to PLOS ONE. After careful consideration, we feel that it has merit but does not fully meet PLOS ONE’s publication criteria as it currently stands. Therefore, we invite you to submit a revised version of the manuscript that addresses the points raised during the review process.

 A sincere apologies to the authors on the delayed review process. I had a total of 14 reviewer rejections with a single reviewer feedback. I have made a decision on this manuscript based on the feedback of that reviewer and also provided mine below.

We look forward to receiving your revised manuscript.

Kind regards,

Bernard X W Liew

Academic Editor

PLOS ONE

Journal Requirements:

“I have read the journal's policy and the authors of this manuscript have the following competing interests:

The INCUS NOVA wearable device used in the research article is a license product of INCUS Performance Ltd. CPS, EF, LW, EF, DP, and HL were employees of INCUS Performance Ltd. at the time the research was completed.”

Additional Editor Comments:

Comments

1. Please state the hypotheses at the end of the Introduction

2. How did you ensure time synchronisation of all devices? If not, what mechanism was performed to ensure variables from the same time-instance were compared across devices

3. Each subject had 5 running speeds. Apparently you combined the observations of all subjects and speeds and performed ICC. Have I misinterpreted this? If you have done it, it is incorrect, given that a standard ICC requires individual observations to be independent. Once you have repeated measures, there is variation within each subject that is not accounted for.

4. please explicitly state the bonferroni correction.

5. provide confidence interval for estimates.

6. I am unsure why a comparison between IMU devices are needed. If the aim is to compare which IMU devices is the best, you simply need to compare it with a benchmark. Provide a 95% confidence interval of the error with the benchmark, and that should suffice.

7. I think the authors need to think about is the differences in reliability/validity of each device confounded by different locations? Is it a device issue or a location issue, or likely a mixture of both?

Reviewers' comments:

Reviewer's Responses to Questions

**Comments to the Author**

1. Is the manuscript technically sound, and do the data support the conclusions?

Reviewer #1: Yes

2. Has the statistical analysis been performed appropriately and rigorously? 

Reviewer #1: Yes

3. Have the authors made all data underlying the findings in their manuscript fully available?

Reviewer #1: Yes

4. Is the manuscript presented in an intelligible fashion and written in standard English?

Reviewer #1: Yes

5. Review Comments to the Author

Reviewer #1: Thank you for the opportunity to review this manuscript. The manuscript's objectives were to determine the validity and reliability of the INCUS NOVA device compared to certain wearable devices (Garmin Running Dynamics Pod, Garmin HR Monitor, and Stride Running Pod) for measuring vertical oscillation in comparison to 2d video measurements. The results indicate that the INCUS NOVA is an acceptable device for measuring vertical oscillation and are comparable to other devices on the market, with some smaller overestimations and underestimations. I would like to commend the authors for a paper that was clear, well-written, and had all the necessary information for a validation study. The findings of this study may only apply to those using the INCUS NOVA (and the manufacturers for making claims about the device), but the methods are also well presented to allow future research to replicate the details for a validation study. My specific comments are listed below:

Line 47: Since you mention that wearable devices provide feedback with "a variety of physiological and biomechanical information", it would be good to include some physiological metrics on top of speed and cadence. OR since your paper is primarily focused on biomechanics, you can exclude physiology and just focus on biomechanics.

Line 56: Some examples of visual and auditory feedback would help the reader better understand your claim.

Line 60: It should be inertial measurement unit.

Lines 75-80: I think the objective statement can flow a little better from the introduction. This paper is clearly aimed towards validating the INCUS NOVA, yet on lines 71-72, you mentioned that the validity and reliability of VO for RDP and Stryd have yet to be validated as well. I recommend either broadening the objective to include the validation of these other devices, or have a smoother transition into the objective that focuses only on INCUS.

Line 105 (and throughout): VMCS is not a standard abbreviation and is confusing. Since you don't use it often, I recommend writing out the full term.

Lines 263-264: I agree that, depending on the device, caution should be taken when interpreting the values. I recommend adding in the discussion what some potential consequences are with overestimating and underestimating VO in terms of performance and injury.

Figures: All of these figures need to be improved, in my opinion. They are very low quality and the plots don't even have axes.

6. PLOS authors have the option to publish the peer review history of their article (what does this mean?). If published, this will include your full peer review and any attached files.

Reviewer #1: **Yes: **Christian Clermont

---

## [Author Response · Author response to Decision Letter 0]

30 Sep 2022

We have responded to the editor and reviewer comments in the 'Response to Reviewers' document, and revised the manuscript accordingly with a copy of the manuscript with and without track changes submitted. We have also updated our Competing Interests statement within our cover letter as requested.

---

## [Editor Report · Decision Letter 1]

10 Oct 2022

PONE-D-22-11731R1The Validity and Reliability of Wearable Devices for the Measurement of Vertical Oscillation for Running.PLOS ONE

Dear Dr. Smith,

Thank you for submitting your manuscript to PLOS ONE. After careful consideration, we feel that it has merit but does not fully meet PLOS ONE’s publication criteria as it currently stands. Therefore, we invite you to submit a revised version of the manuscript that addresses the points raised during the review process.

We look forward to receiving your revised manuscript.

Kind regards,

Bernard X W Liew

Academic Editor

PLOS ONE

Journal Requirements:

Additional Editor Comments:

There are some minor points to be corrected prior to acceptance. I do not think the corrections to my prior comments were addressed adequately.

1. Hypothesis. Can you please state a hypothesis in the form of which method would you think is the best, based on prior research or theory. A directional hypothesis.

2. Please stated the corrected alpha when stating bonferroni correction - e.g. 0.05/5

3. Can you explicitly mention the term time synchronisation ins the paragraph starting with lin 147? E.g. time normalisation was achieved by.....

---

## [Author Response · Author response to Decision Letter 1]

13 Oct 2022

We have addressed and responded to the Editor comments in the 'response to reviewers' document.

---

## [Editor Report · Decision Letter 2]

18 Oct 2022

PONE-D-22-11731R2The Validity and Reliability of Wearable Devices for the Measurement of Vertical Oscillation for Running.PLOS ONE

Dear Dr. Smith,

Thank you for submitting your manuscript to PLOS ONE. After careful consideration, we feel that it has merit but does not fully meet PLOS ONE’s publication criteria as it currently stands. Therefore, we invite you to submit a revised version of the manuscript that addresses the points raised during the review process.

We look forward to receiving your revised manuscript.

Kind regards,

Bernard X W Liew

Academic Editor

PLOS ONE

Journal Requirements:

Additional Editor Comments:

Dear authors,

Before I can accept, there is a mistake in your response. For the Bonferroni correction, you mentioned " the probability values for the paired T-tests were Bonferroni corrected, therefore multiplied by the number of comparisons (i.e. p*10)." If the alpha is 0.05, and it is 0.5, than the correction makes things more lenient. It should be divide. I hope this was a typographical mistake rather than a statistical mistake. Can you clarify?

Regards,

Bernard

---

## [Author Response · Author response to Decision Letter 2]

2 Nov 2022

The response to reviewers is included in the Response to Reviewers Document.

---

## [Editor Report · Decision Letter 3]

4 Nov 2022

The Validity and Reliability of Wearable Devices for the Measurement of Vertical Oscillation for Running.

PONE-D-22-11731R3

Dear Dr. Smith,

We’re pleased to inform you that your manuscript has been judged scientifically suitable for publication and will be formally accepted for publication once it meets all outstanding technical requirements.

Kind regards,

Bernard X W Liew

Academic Editor

PLOS ONE